# The evolution of interdisciplinarity and internationalization in scientific journals

Huaxia Zhou[1], Luis A Nunes Amaral[2,3,4,5]*

[1]Department of Electrical and Computer Engineering, Northwestern University, Evanston, United States; [2]Department of Engineering Sciences and Applied Mathematics, Northwestern University, Evanston, United States; [3]Department of Physics and Astronomy, Northwestern University, Evanston, United States; [4]Northwestern Institute on Complex Systems (NICO), Northwestern University, Evanston, United States; [5]NSF-Simons National Institute for Theory and Mathematics in Biology (NITMB), Chicago, United States

## eLife Assessment

This **important** study uses data from OpenAlex on more than 50 million journal articles in over 50,000 research journals to examine the dynamics of interdisciplinarity and international collaboration in research journals. The data analytics used to quantify disciplinary and national diversity are **convincing**, and support the claims that journals have become more diverse in both aspects. The revisions made by the authors have addressed the small number of concerns the reviewers had about the original version.

*For correspondence:
amaral@northwestern.edu

Competing interest: The authors declare that no competing interests exist.

**Abstract** There is a widely held perception that science is becoming more international—through multi-national collaborations—and interdisciplinary, drawing on knowledge from multiple domains. However, these hypothesized trends have not yet been quantitatively characterized. With the publication metadata from OpenAlex, we examine trends in two groups of journals: disciplinary journals in natural sciences, life sciences, social sciences, and multidisciplinary journals that publish articles in multiple fields. Supporting existing perceptions, we find an almost universal trend towards increasing internationalization of both sets of journals. Nevertheless, we find disparities: medicine journals are less international than journals in other disciplines and do not increase their levels of internationalization, whereas physics journals appear to be segregating between those that are international and those that are not. We also find that multidisciplinary journals have undergone significant shifts in their disciplinary focuses over the past century, whereas disciplinary journals appear to have largely maintained their degree of interdisciplinarity.

## Introduction

Peer-reviewed scientific journals are the backbone of scholarly communication for the worldwide community of scholars (*Russell, 2001*). Remarkably, the volume of publications in scholarly journals has been growing exponentially for over a century (See *Figure 1b*; *de Solla Price, 1986*). The dominance of journal articles was not always the state of affairs in scientific communication. The longest-running scholarly journal was launched in 1665 (the French *Le Journal des Sçavans* [or *Journal des savans*] was published for the first time a few months earlier but its publication was interrupted twice *Partridge, 2015*). — the Royal Society of London launched the *Philosophical Transactions of the Royal Society* to collect and distribute important scientific findings and news (*Fyfe et al., 2022*). Unlike nowadays, in the 18th and 19th centuries, monographs, literary essays, public and private presentations, personal

**Figure 1.** Scholarly journal articles are the largest category of scholarly publications. The number of journal publications increased steadily except during the two World Wars. Solid lines show the fits to the data which have, in some cases, been shifted vertically for clarity. (**a**) Scholarly works (with relevant metadata) indexed in the OpenAlex database are categorized into seven groups. Most publications are journal articles. (**b**) Number of journal articles has been doubling approximately every 10 years for over a century. While there were significant drops during World War I (1914 to 1918) and World War II (1939 to 1945), the growth of journal article publications quickly recovered. The solid line shows a linear fit to the logarithm of the dependent variable. (**c**) Number of actively publishing journals has been doubling approximately every 14 years for over a century, but the growth rate has not been constant. The two solid lines show the linear fits to the logarithm of the dependent variable to different time periods. (**d**) Number of countries represented in authors' affiliations for journal articles published each year. The data is well described by a logistic curve centered at 1970. Most of the growth in the number of countries occurred between 1949 and 1991.

letters, pamphlets, and full-length books were all significant channels for communicating research findings and established priority (***Baldwin, 2020***). By the turn of the 20th century, printed scientific journals emerged as the primary means of disseminating new scientific discoveries (***Baldwin, 2020***; ***Friedlander and Bessette, 2003***). In this new millennium, digital publications, still within the framework of scholarly journals, have overtaken printed publications (***Tomlin, 2005***).

The scope of scientific enterprise has also evolved over time. In the 380 s B.C.E., Aristotle classified all human knowledge into two philosophical categories: speculative and practical (***Barnes, 2000***). Practical philosophy encompassed topics, such as economics, ethics, and politics. Speculative philosophy encompassed aesthetics, mathematics, metaphysics, and physics (also called natural philosophy) (***Grant, 2007***). Not until the 19th century did the current partitioning of scholarly endeavors start to take shape (***Cahan, 2003***).

This re-arrangement of human knowledge was accompanied by its exponential growth, prompting increasing specialization of scholars and the emergence of ever more specialized venues for sharing discoveries. Perhaps as a result of this specialization trend, the number of scholarly journals has been doubling approximately every 14 years (***Figure 1c*** and Appendix 1). Much of this growth has occurred

within the dominant natural sciences — biology, chemistry, and physics — each now encompassing a multitude of subfields (*Casadevall and Fang, 2014*, *Wagner et al., 2011*, *van Raan, 1999*). Counter-intuitively, increasing specialization has produced a drive for interdisciplinarity, with researchers increasingly collaborating to exchange knowledge and methods across domains and bridge disciplinary boundaries (*Silva et al., 2013*, *Abbott, 2010*, *Wray, 2005*, *Dogan and Pahre, 1990*). Due to this drive, there had been a second trend in opposition to ever more specialized journals: the increasing prominence of multidisciplinary journals (e.g. *Nature*, *Science*, and *PNAS*), which feature research contributions from multiple disciplines (*Silva et al., 2013*, *van Raan, 1999*).

Another factor that has transformed the worldwide scientific enterprise is its increasing internationalization, with scientific collaborations now extending across national boundaries. Whereas France, Germany, the United Kingdom (U.K.), and the United States (U.S.) dominated knowledge production in the 19th and 20th centuries, in recent decades, we have witnessed the increasing importance to knowledge creation of Asian nations, such as China, India, and Japan (*Appendix 2—figure 1*).

In fact, there has been a significant expansion in the number of countries where scientists are publishing in selective journals. As seen in *Figure 1d*, this expansion is well described by a logistic model with a growth rate that peaked in 1970, but extends over the period 1949–1991. Plausibly, the widespread focus on increased scientific research was due to the importance of science and technology to the outcomes of the World Wars.

Recent literature has highlighted the importance of team science in the production of knowledge, especially impactful knowledge, in recent decades. A less well-understood aspect in the growth of team science is the impact of interdisciplinarity and internationalization. A driving thrust of this study is the assumption that perceived trends towards greater interdisciplinarity and internationalization will be visible and quantifiable using data-driven approaches. We further hypothesize that quantifying these processes will yield a deeper understanding of the observed changes. These insights will guide researchers and policy makers in promoting collaborations among different disciplines and countries, thus encouraging the type of innovative research necessary to address the complex challenges currently faced by humanity.

## Results
### Top multidisciplinary journals

To build confidence in our metrics, we start by investigating the interdisciplinarity and internationalization of three of the most highly regarded multidisciplinary journals — *Nature*, *Science*, and *The Proceedings of the National Academy of Sciences of the United States of America* (*PNAS*). All three journals have a long history and publish a large number of articles annually, mostly original research.

*Figure 2a* shows the temporal evolution of the interdisciplinarity index for these three journals. It is visually apparent that *Nature* and *PNAS*, in particular, have experienced significant shifts in the degree of interdisciplinarity of the articles they publish. To better understand the specifics of these changes, we next explore how different disciplines received evolving degrees of attention in each of these journals over time (*Figure 2b-d*). Because of their preponderance in the data, we split Biology, Chemistry, and Physics articles into separate groups and pooled all articles on other major fields of study into an 'Other Fields' group.

Upon this breakdown, a clear trend emerges across all three journals. An initial period where a broad range of disciplines are published in those journals, followed by a transition to a greater emphasis on Biology, Chemistry, and Physics. Specifically, in the 1920s, nearly 60% of articles in *Nature* and *Science* concerned disciplines other than Biology, Chemistry, or Physics (*Figure 2c and d*). By 1990, approximately 90% of articles in *PNAS* reported on studies in either Biology or Chemistry (*Figure 2b*). For *Nature*, the peak concentration was reached in the 1970 s, when 60% of articles reported on studies in either Biology or Chemistry. Despite distinct historical trajectories, all three journals have evolved in recent years toward a situation where nearly 40% of the articles they publish fall outside of the dominant disciplines, but where Biology studies account for 20–30% of all publications.

*Figure 3a* shows the time evolution of the internationalization index for *Nature*, *PNAS*, and *Science*. It is visually apparent that *PNAS* and *Science*, in particular, have experienced an extraordinary growth in the diversity of countries from which authors of published articles originate. To better understand the specifics of these aggregate trends, we next explore how authors affiliated with institutions

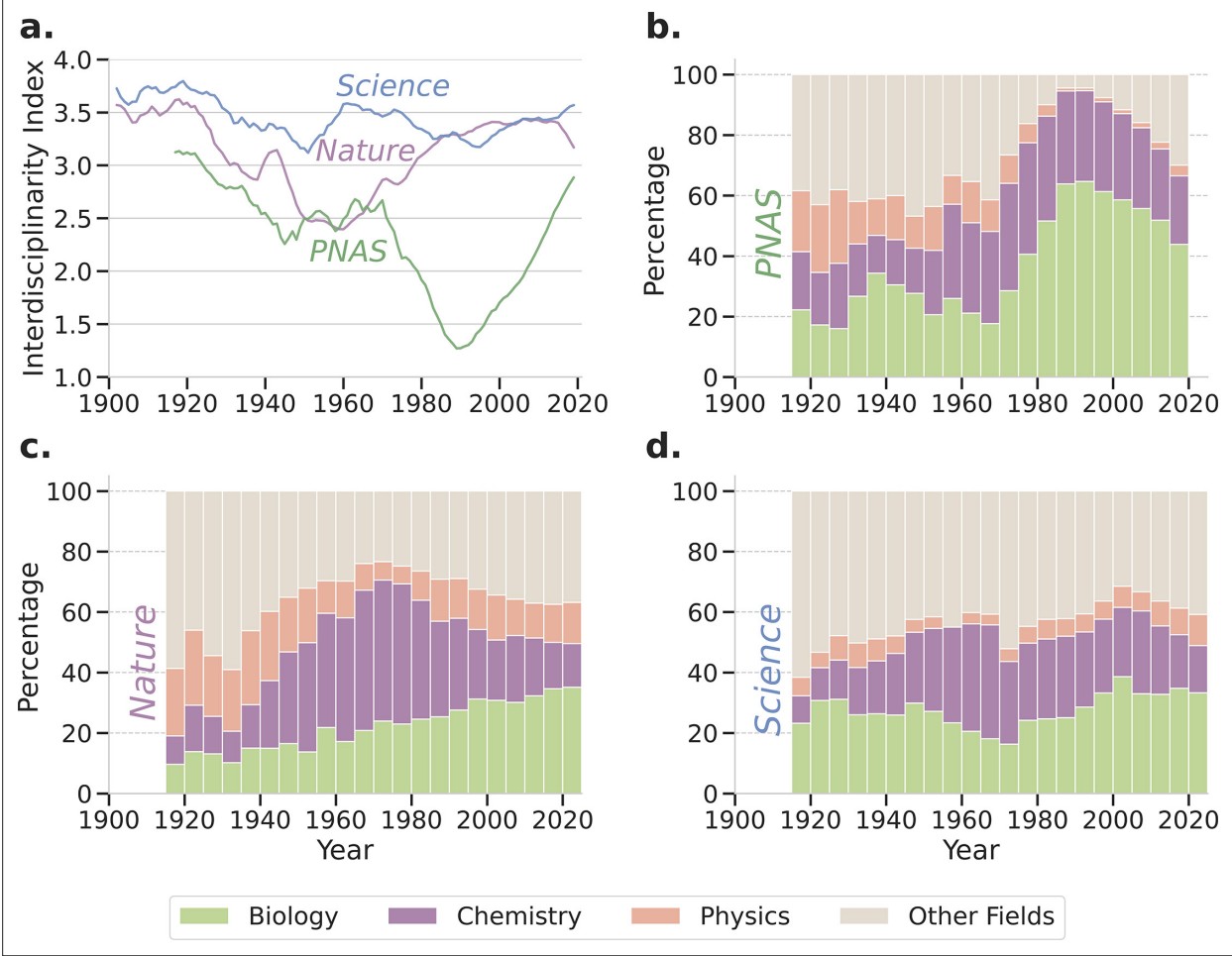

**Figure 2.** Long-standing, high-reputation multidisciplinary journals have shifted their disciplinary focus quite dramatically over the last 100 years. (**a**) Time evolution of the interdisciplinarity indices, $I_d$, of *Nature*, *Science*, and *PNAS*, measured by the entropy of the annual distribution of disciplines in published articles. (**b-d**) Time evolution of the disciplinary partitioning of articles published in *PNAS* (**b**), *Nature* (**c**), and *Science* (**d**). For all three journals, there has been a shift from 'Other Fields' to Physics, Chemistry and Biology. In the 1920s, nearly 60% of articles in *Nature* or *Science* were in fields other than Physics, Chemistry, and Biology. Interestingly, after following different paths, all three journals are now publishing nearly 40% of their articles in fields other than Physics, Chemistry, and Biology. In *PNAS*, there was a noticeable shift towards Biology from 1960 to 1990 and away from Physics and 'Other Fields.' After 1990, 'Other Fields' grew at the expense of Biology. In *Nature*, there has been a steady growth of Biology-focused articles. For both *Nature* or *Science*, the number of Chemistry publications began its ascent in 1940, reaching a pinnacle around 1970, and a slower decline after that.

located in countries from different regions were increasingly able to publish in these leading multi-disciplinary journals (*Figure 3b-d*). Because of their preponderance in the data, we group countries into three main regions: Asia, Europe, and North America. We group all affiliations located outside of those regions into 'Other Regions'.

Upon this breakdown, a clear trend emerges for the two U.S.-based journals. Both *PNAS* and *Science* started as a publication outlet for only authors with affiliations in North American institutions (*Figure 3b and d*). Starting in the 1980s, both experienced an increase in the fraction of papers authored by researchers affiliated with European institutions and, to a lesser extent, Asian institutions. Currently, about 60% of papers published in *PNAS* and *Science* are authored by researchers affiliated with North American institutions, but that percentage appears likely to continue to decline.

The story is more complex for *Nature* (*Figure 3c*). Already over a century ago, authors from institutions in Europe and North America were publishing in this U.K.-based journal. Moreover, already by the 1940 s, authors affiliated with institutions in Asia started publishing in *Nature*. Despite this early internationalization effort, *Nature* now publishes about the same fractions of Asian, European and North American authors as *Science*, but a smaller fraction of authors affiliated with institutions from Asia than *PNAS*.

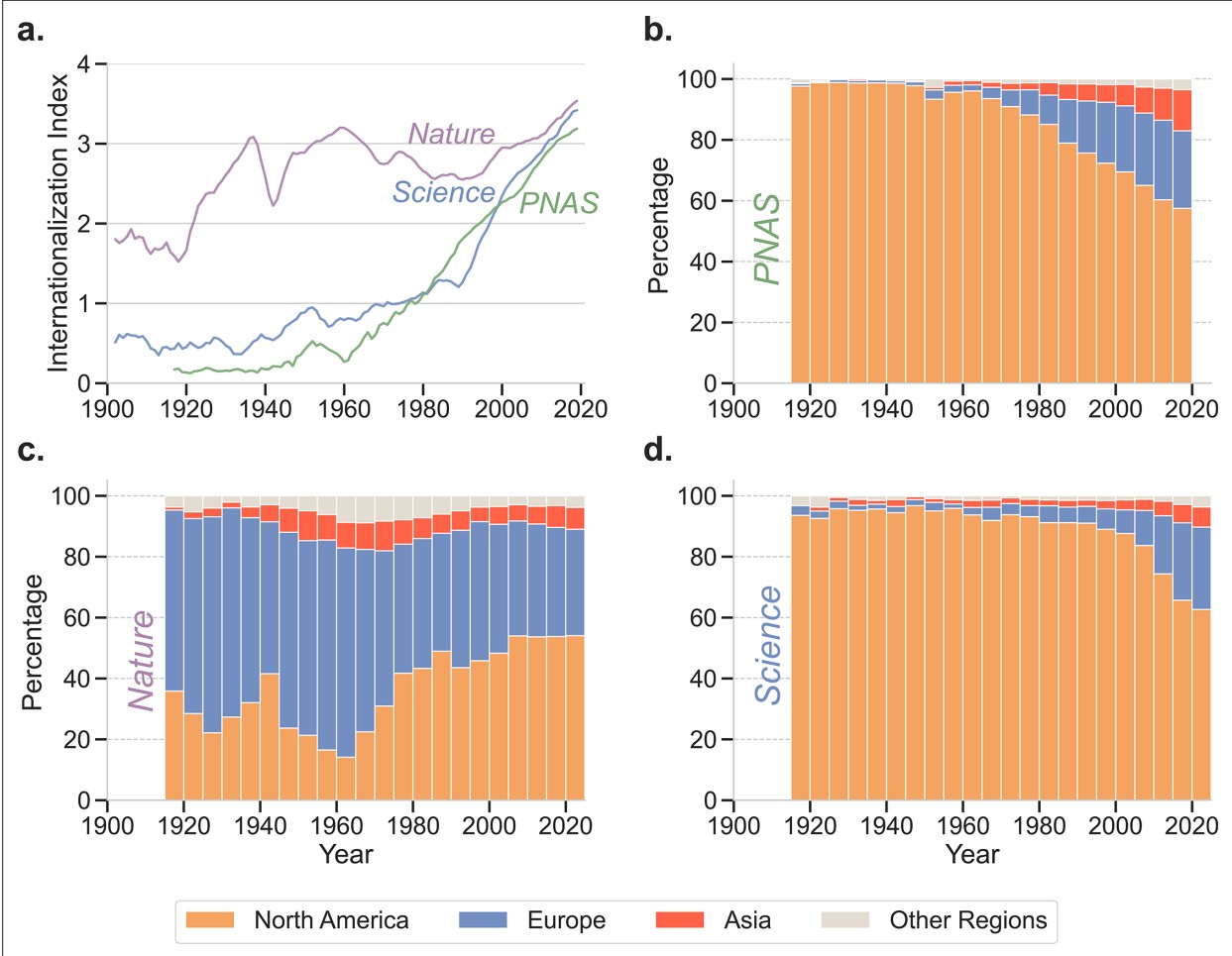

**Figure 3.** Long-standing, high-reputation multidisciplinary journals have become more international. (**a**) Time evolution of the internationalization indices, $I_n$, of *Nature*, *Science*, and *PNAS*, measured by the entropy of location of the affiliations associated with the authors of published articles. The index has increased drastically over the last 100 years for all three journals. (**b-d**) Time evolution of the percentage of articles whose authors are affiliated with institutions from Asia, Europe, or North America in *PNAS* (**b**), *Nature* (**c**), and *Science* (**d**). For all three journals, the greatest growth has occurred for authors with affiliations in institutions located in Asia. Surprisingly, *PNAS* now has the largest fraction of authors from Asian institutions, even though *Nature* had a higher level in the 1940s. For both *Science* and *PNAS*, the fraction of authors from European institutions has increased dramatically since the 1960s. Around the same time, the fraction of authors from European institutions publishing in *Nature* decreased dramatically as the fraction of authors from North American institutions grew rapidly.

This internationalization shift may be partially attributed to external factors, such as the rise of electronic publishing which now offers researchers a multitude of platforms for communicating their research and an easier path to submitting manuscripts to — or reviewing manuscripts for — journals based in other continents. Electronic publishing also enabled a new model — open access (*Suiter and Sarli, 2019*, *Tomlin, 2005*). Initial studies suggested that open access publication produced greater research visibility and greater number of citations (*Craig et al., 2007*). Another important factor impacting journal internationalization is the growth of team sizes (*Duch et al., 2010*; *Guimerà et al., 2005*) and of international collaborations as more diverse collaborating teams may prioritize different factors in the choice of publication venue.

## Disciplinary and multidisciplinary journals: Interdisciplinarity

Next, we compare the changes in interdisciplinarity and internationalization for disciplinary journals (see Materials and methods for details). Reassuringly, our analysis shows that over the observed time periods, the interdisciplinarity index ($I_d$) of multidisciplinary journals consistently remains significantly higher than that of disciplinary journals (*Figure 4* and *Appendix 6—figure 2*). Considering just the most recent indices (as shown in *Table 1*), we found that the average interdisciplinarity index for

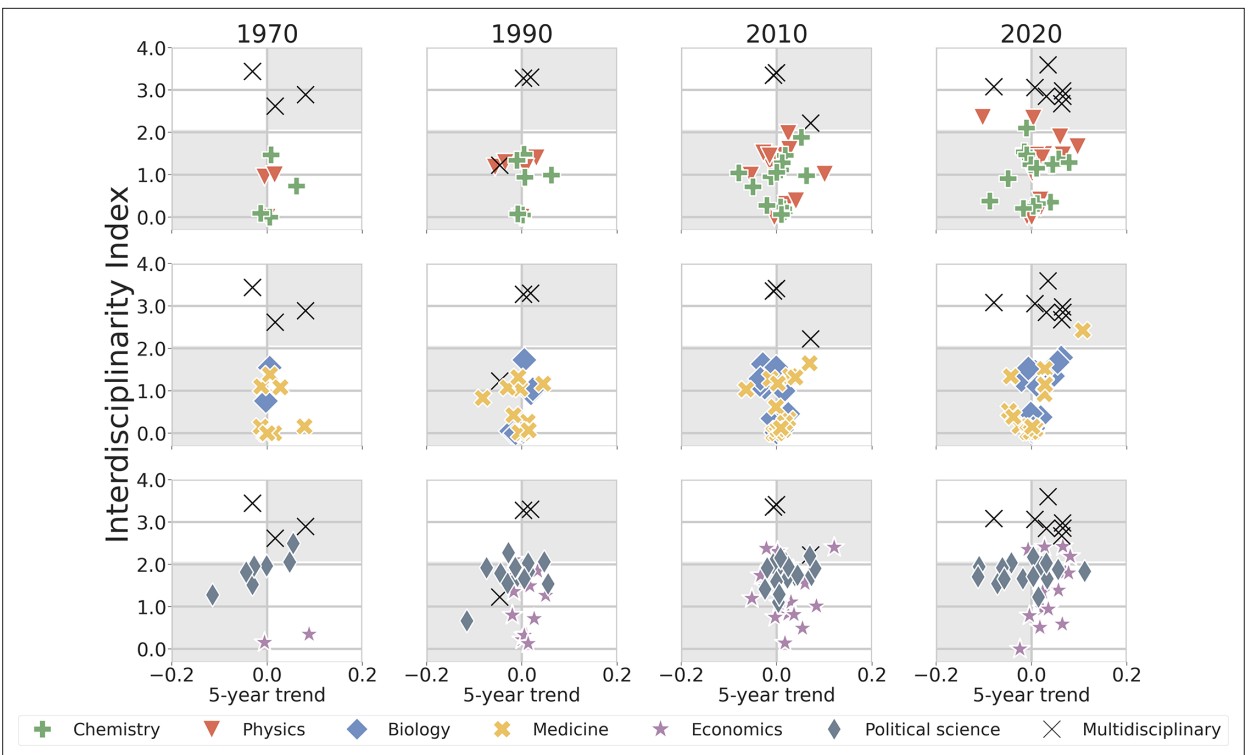

**Figure 4.** Trends in interdisciplinarity across journals in natural sciences, life sciences, social sciences, and multidisciplinary sciences in 1970, 1990, 2010, and 2020. Most recent interdisciplinarity index at a specific year, $I_d$, versus the linear trend of the journal's interdisciplinarity index over the preceding 5 years. Multidisciplinary journals consistently exhibit a substantially higher interdisciplinarity index ($I_d \approx 3$) compared to all disciplinary journals throughout the observed period (**Table 1**). Disciplinary journals in natural sciences (Chemistry and Physics) and life sciences (Biology and Medicine) appear to fall into two groups based on their interdisciplinarity indices: The first group of journals maintains a narrow focus, publishing research that draws on a single discipline ($I_d \approx 0$). The second group of journals publishes moderately interdisciplinary research ($I_d \approx 1.5$). This separation between mono-disciplinary and moderately interdisciplinary journals is visually apparent for journals in natural and life sciences. In contrast, the disciplinary journals in Economics and Political Science we studied seem to be moderately interdisciplinary ($I_d \approx 1.5$).

multidisciplinary journals is approximately 3 (*Science* has the highest interdisciplinarity index, $I_d \approx 3.5$). Moreover, the interdisciplinarity of multidisciplinary journals appears to reach a steady state after 1990.

The evolution of interdisciplinarity is more diverse for disciplinary journals. Physics and Chemistry journals display intriguing changes in interdisciplinarity. A majority of those journals has increased dramatically in interdisciplinarity over time, with Physics journals, in particular, reaching very high degrees of interdisciplinarity. In contrast, a smaller subset of journals have become very

**Table 1.** Average interdisciplinarity characteristics of journals from different disciplines.
We report the average Interdisciplinarity Index $I_d$, its most recent 5 year trend, the Spearman's correlation coefficient of $I_d$ and the most recent 5 year trend, and the p-value of the correlation. We highlight statistically significant cases in bold face.

| Discipline | Mean $I_d$ | Mean 5 yr trend | $\rho$ | p-value |
|---|---|---|---|---|
| Biology | 0.86 | 0.01 | 0.56 | 0.02 |
| Chemistry | 1.04 | 0.02 | 0.28 | 0.30 |
| Physics | 1.14 | 0.01 | 0.32 | 0.24 |
| Economics | 1.53 | 0.04 | 0.36 | 0.16 |
| Medicine | 0.68 | –0.01 | 0.15 | 0.58 |
| Political Science | 1.78 | –0.01 | 0.06 | 0.83 |
| Multidisciplinary | 3.02 | 0.02 | –0.75 | 0.05 |

mono-disciplinary. Consistent with this finding, we categorize disciplinary journals into two groups according to their interdisciplinarity index: mono-disciplinary ($I_d \approx 0$) and moderately interdisciplinary ($I_d \approx 1.5$).

A segregation between moderately interdisciplinary and monodisciplinary journals is not as apparent for Biology and Medicine journals. The small values of the 5 year trends show that the interdisciplinarity of Biology and Medicine journals is also more stable than that of Physics and Chemistry journals.

Economics journals show a trend similar to that of the journals in the natural and life sciences. That is, their interdisciplinarity has increased over time and there is a spread in the levels of interdisciplinarity around moderate interdisciplinarity. In contrast, Political Science journals have maintained high levels of interdisciplinarity for the entire period studied — a level that was achieved only recently by the most interdisciplinary Physics journals. Perhaps this is not surprisingly as political science is a multifaceted discipline that incorporates insights from fields as diverse as business, economics, law, psychology, and sociology (*de Bakker et al., 2019*).

Thus, despite there being an overall perception of a growing emphasis on interdisciplinary research, the reality is that some disciplinary journals have strengthened their mono-disciplinary focus. The reasons for this are likely complex and multifaceted but may be, at least in part, due to traditional academic structures and publishing norms that favored depth over breadth. Indeed, some of the mono-disciplinary journals in Biology, Chemistry and Physics are highly regarded journals — *Molecular Cell* ($I_d \approx 0.03$), *Journal of the American Chemical Society* ($I_d \approx 0.3$), and *Nature Physics* ($I_d \approx 0.3$). However, we also observe a statistically significant positive correlation between the latest value of $I_d$ and its most recent 5 year trend for Biology journals (*Table 1*), suggesting that Biology journals which are already interdisciplinary are continuing to increase their degree of interdisciplinarity. The

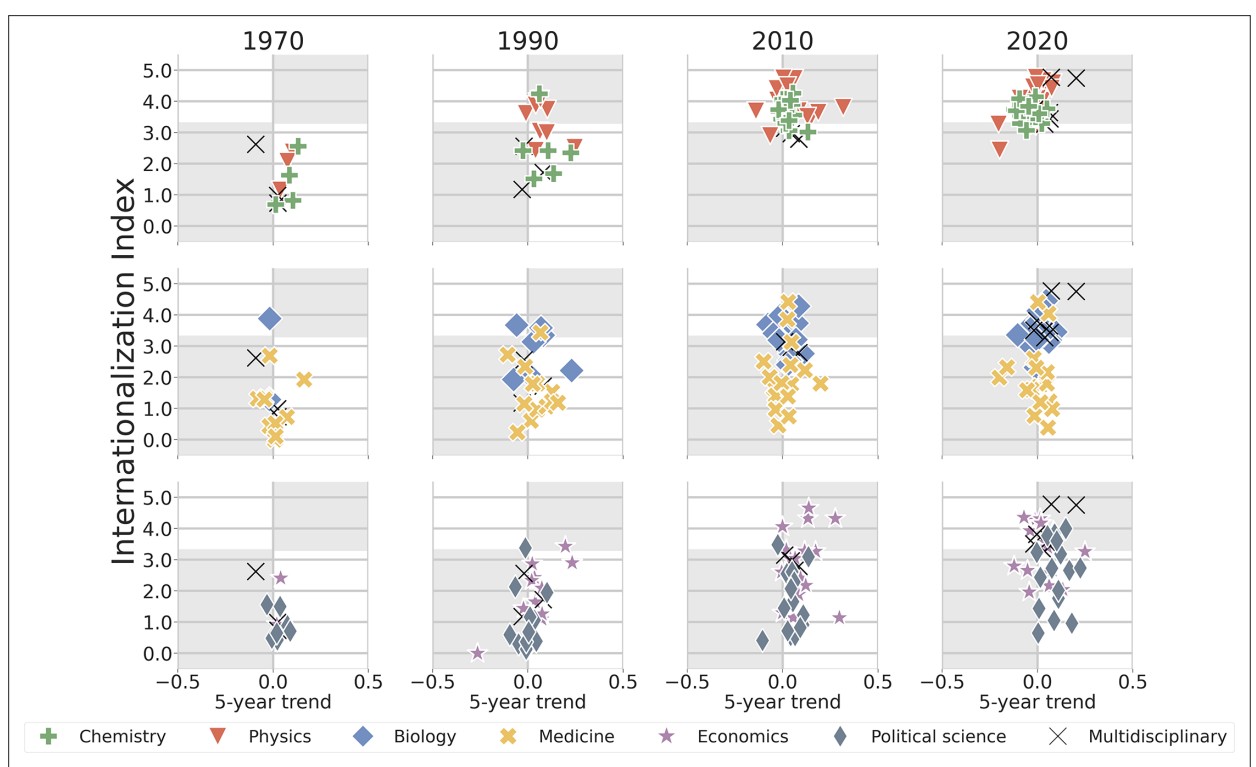

**Figure 5.** Trends in internationalization across journals in natural sciences, life sciences, social sciences, and multidisciplinary sciences in 1970, 1990, 2010, and 2020. Most recent internationalization index at a specific year, $I_n$, versus the linear trend of the journal's internationalization index over the preceding 5 years. The internationalization of disciplinary journals exhibits a variety of behaviors (see also *Table 2*). Physics journals have steadily increased their internationalization, which is now higher than for most multidisciplinary journals, comparable to those of multidisciplinary journals during this period. In 2020, the mean $I_n$ of Physics journals is approximately 4.1. Journals in Biology and Chemistry display similar trends, but with lower mean $I_n$, approximately 3.6 for both Chemistry and Biology journals. In contrast, while a few Medicine and Economics journals have $I_n \approx 4$, most journals in these disciplines and in Political Science have $I_n < 3$.

**Table 2.** Average internationalization characteristics of journals from different disciplines.
We report the average Internationalization Index $I_n$, its most recent 5 year trend, the Spearman's correlation coefficient of $I_n$ and the most recent 5 year trend, and the p-value of the correlation. We highlight statistically significant cases in bold face.

| Discipline | Mean $I_n$ | Mean 5 yr trend | $\rho$ | p-value |
|---|---|---|---|---|
| Biology | 3.58 | 0.02 | 0.18 | 0.51 |
| Chemistry | 3.61 | –0.03 | –0.06 | 0.82 |
| Physics | 4.08 | –0.02 | 0.54 | 0.03 |
| Economics | 3.24 | 0.01 | 0.16 | 0.54 |
| Medicine | 1.94 | –0.01 | –0.19 | 0.47 |
| Political Science | 2.68 | 0.11 | 0.17 | 0.53 |
| Multidisciplinary | 3.90 | 0.05 | 0.36 | 0.43 |

emergence of novel areas, such as Systems and Synthetic Biology may be responsible for this trend (*Friboulet and Thomas, 2005*).

### Disciplinary and multidisciplinary journals: Internationalization

We next compare the internationalization index of disciplinary and multidisciplinary journals over time (*Figure 5* and *Appendix 7—figure 2*). With regard to internationalization, multidisciplinary journals are not the trend-setters (*Table 2*). Instead, disciplinary journals in Physics have the highest $I_n$, while disciplinary journals in Biology and Chemistry have internationalization indices comparable to that of multidisciplinary journals. An exception to this pattern is the pair of mega open-access journals — *PLOS One* and *Scientific Reports* — which have internationalization indices as high as those of the most internationalized Physics journals. The remaining five multidisciplinary journals exhibit internationalization indices comparable to those of Biology and Chemistry journals. This finding is perhaps unsurprising, given the substantial representation of biological and chemical research within these multidisciplinary journals.

The very high degree of internationalization of Physics journals is also not surprising. Large international initiatives, such as the Conseil Européen pour la Recherche Nucléaire (CERN) colliders (*CERN, 2019*), the Super-Kamiokaze neutrino detection facility (*Fukuda et al., 2003*), the Laser Interferometer Gravitational-Wave Observatory (LIGO) (*Barish and Weiss, 1999*) and the Paranal Observatory (or Observatorio Paranal) *Dekker et al., 2000* have been a constant of the Physics research landscape since the 1950s.

Our finding also reveals a statistically significant correlation between the latest value of $I_n$ and its most recent 5 year trend for disciplinary journals in Physics. This correlation is consistent with the idea that Physics journals that are already highly internationalized are becoming even more internationalized, while the least internationalized are becoming less internationalized. This trend highlights a growing divide between journals actively pursuing internationalization and those either unwilling or unable to do so.

Similar to physical research, biological research has been revolutionized by global initiatives, such as the Human Genome Project (HGP) (*Collins et al., 1998*), the International HapMap Project (*Gibbs et al., 2003*), or the 1000 Genomes Project (1KGP) (*Siva, 2008*). Remarkably, disciplinary journals in Chemistry have achieved a similarly high degree of internationalization without the benefit of large international collaborations. The explanation here may be the global need for the spread of the fossil fuels that are so critical to the chemical industry and have promoted development of the discipline across the globe (*Scott et al., 2015*).

Among disciplinary journals, Medicine journals exhibit the lowest level of internationalization (mean $I_n \approx 1.9$). This finding aligns with prior reports that highlight the relatively lower levels of international collaboration in applied and clinical medicine (*Davidson Frame and Carpenter, 1979*). There are, however, two notable exceptions: *The Lancet* ($I_n \approx 4.0$) and the *European Journal of Clinical Investigation* ($I_n \approx 4.4$). Since most medicine journals maintain a modestly low internationalization index, we surmise that medical research adopts a predominantly regional focus and field-specific norms, such as localized health issues, regional regulatory standards, etc. Nonetheless, it is likely that

**Table 3.** Average team size of journal publications in different disciplines.
We report the average team size and its standard deviation for publications across various disciplines. We highlight Physics in bold face since it stands out for having the largest average team size.

| Discipline | Mean team size | $\sigma$ |
|---|---|---|
| Biology | 4.84 | 4.96 |
| Chemistry | 5.02 | 3.12 |
| Physics | 8.52 | 75.48 |
| Economics | 1.95 | 1.21 |
| Medicine | 3.31 | 6.77 |
| Political Science | 1.36 | 0.85 |
| Multidisciplinary | 5.40 | 6.89 |

global epidemics — Acquired immunodeficiency syndrome (AIDS), Severe acute respiratory syndrome (SARS), Middle East respiratory syndrome (MERS), or Coronavirus disease 2019 (COVID-19) — will increase the pressure to internationalize medical research.

The need for an increase in the internationalization of social sciences research may be less obvious. Indeed, even some Political Science have low internationalization (mean $I_n \approx 2.7$). This may be due to differences in what types of questions researchers from different countries may want to investigate. However, there has been a recent interest by researchers in the Western world in investigating phenomena that extend beyond what is expected from Western, educated, industrialized, rich and democratic (WEIRD) societies.

## Disciplinary journals: Impact of team size

Since very large collaborations are not unusual in Biology, Medicine, or Physics (*Table 3*), we investigated whether our findings for those two disciplines could have been due to the characteristics of the largest teams. As a control, we repeated the previous analyses for these three disciplines but excluding all publications with 10 or more authors.

Consistent with other studies, we found that there has been a consistent and significant rise in the number of publications authored by teams of 10+ individuals. This increase was already visible in Physics and Medicine by the 1960s. For Biology, the increase becomes noticeable after 1980 (*Appendix 8—figure 1*).

Our control analyses confirm the results obtained when considering all teams, thus demonstrating that the trends in interdisciplinarity and internationalization are not explainable by the presence of very large teams in those disciplines (*Appendix 8—figure 2*). That is, the observed internationalization of the research published in these top disciplinary journals occurs because the disciplines have created a truly international enterprise.

## Discussion

Our findings highlight a transformational shift within the landscape of scientific publishing — the increasingly interconnected nature of scientific research both at the international and interdisciplinary levels. The reasons for this increased interconnectedness are multifaceted. Increased journal competition for the most exciting research, the need for access to scarce or unique resources, the desire to attract a talented workforce. Whatever the reasons, it is critical to recognize the need for such interconnectedness if one aims to tackle the enormous societal challenges that face us, from climate change to increased conflict.

While this shift may bring hope to some, it may be seen as threatening by others. Either because of a misguided focus on competition between nations and the need to retain claims of primacy, or because of wishes to preserve disciplinary purity. Indeed, some current efforts in some countries to 'safeguard their national research enterprises' may prove costly to the creation of new knowledge within their own nations. Whatever the underlying motivation, it would be wise for policymakers,

research funding organizations, and research institutions to understand and adapt to this shifting landscape in order to effectively promote innovation.

Our study is not without limitations. Perhaps the greatest limitation is that we focus on disciplines where journal articles are the primary scholarly communication channel. This limitation prevents us from being able to generalize our findings to disciplines in the Humanities, which rely on books or book chapters, or in Computer Science and Engineering, which heavily rely on conference proceedings (*Lisée et al., 2008*). A second limitation of our study is the fact that we only consider seven multidisciplinary journals and 16 journals for each discipline. The reason for the small number of multidisciplinary journals considered is that there are not that many highly respected journals thus classified.

Another limitation we must discuss relates to the completeness and representativeness of the data across disciplines. This concern was foremost in our minds when deciding on the time period for analysis. Indeed, the reason why we consider the period 1900–2020 for only the three major interdisciplinary journals — *Nature, PNAS, Science* — has to do precisely with the fact that those journals are the most likely to be accurately covered. When considering disciplinary journals, we restrict our attention to the period 1960–2020. Even then, for some journals and disciplines covered can be spotty prior to 2000. For this reason, we perform analyses that focus on manually selected journals and for specific time points (1970, 1990, 2010, 2020).

Moreover, the selected journals are among the most reputable in their fields and publish large volumes of articles (see Materials and methods for details). While the requirements we impose limit the number of disciplinary journals that we can consider, we believe that the selected journals provide an accurate reflection of the changes occurring within the disciplines we consider.

## Materials and methods

We focused our analysis on the March 2022 snapshot of OpenAlex, a large-scale and open scholarly metadata source. When Microsoft ceased updates of the Microsoft Academic Graph (MAG) at the end of 2021 (*Microsoft Research, 2023*), the non-profit organization OurResearch (*OurResearch, 2024*) incorporated the entire MAG corpus — excluding patent data — into OpenAlex (*Priem et al., 2022*). Hence, the snapshot of OpenAlex we studied maintains all features and data schemas used by the MAG.

The March 2022 snapshot of OpenAlex contains information on 208,755,206 scholarly works. OpenAlex categorizes scholarly works into seven types: journal articles, conference proceedings, repositories, book chapters, books, theses, and datasets (*Figure 1a*). We refined the downloaded data by filtering out entries lacking essential information for our analysis, such as publication year, document type, authorship, affiliation, and field of study information. We then restricted our attention further to scholarly works published after 1900 (due to poor coverage of earlier works) and before 2021 (due to incomplete inclusion of more recent works). This filtering procedure left us with a corpus comprising 56,697,402 publication records, 87% of which are journal articles.

These ~50 million articles were published across 51,062 scholarly journals. Remarkably, *Figure 1b* shows that the number of journal articles published annually has been growing exponentially with an astonishing doubling period of a decade. Even the significant impacts on publication rates of the two World Wars were quickly overcome, and exponential growth resumed.

As journals comprise the largest share of scholarly publications, we choose to study how two characteristics of journal articles — such as diversity of topics and countries — evolved over the last century. A significant challenge with these records, however, is the frequent unavailability of authorship or affiliation metadata — a limitation that also affected the MAG (see *Appendix 3—figure 1*) for a comparison to other databases. The reasons behind this missing metadata in OpenAlex remain unclear. Due to the difficulty of accurately imputing these missing affiliations, our analysis focuses exclusively on journal article publications that include both authorship and affiliation metadata. Approximately 54% of journal articles in the database have complete authorship and affiliation metadata.

### Metadata

Every journal article recorded by OpenAlex is assigned to a unique identifier: 'PaperId,' and the journal where it was published is also assigned to a unique identifier: 'JournalId' (*Priem et al., 2022*). These internal identifiers also connect to external identifiers, such as the International Standard Serial

Number (ISSN), the electronic ISSN (eISSN), and the Research Organization Registry Identifier (ROR ID) (*Research Organization Registry, 2024*), enabling us to connect information across multiple external data sources. In particular, we can connect authors' affiliations to institutions listed in the ROR database (*Research Organization Registry, 2024*) and thus identify the country where the institution is located.

We then referred to the Clarivate Analytics' Journal Citation Reports (Journal Citation Reports can be accessed through https://jcr.clarivate.com/jcr/browse-categories) to obtain the disciplinary classification of the journals in our corpus (*Krampl, 2019*, *Abrizah et al., 2015*). We selected seven multidisciplinary journals and 16 journals for each of six disciplines across the natural sciences (Chemistry and Physics); life sciences (Biology and Medicine); and social sciences (Economics and Political Science) for a detailed investigation. We selected journals with long publishing history, large number of articles published annually, high impact factors (based on the journal impact factor in 2022), and restricted our attention to journals primarily publishing original research, thus excluding journals focusing on reviews. We then obtained the full publication records of each journal from OpenAlex corpus (*Priem et al., 2022*) (see *Appendix 4—figure 1*).

## Quantification

The interdisciplinarity of a journal is measured by the extent to which its articles draw upon knowledge and methods from diverse disciplines. To quantify this, we make use of the fact that we can calculate the probability distribution $p_{jt}(d)$ of the primary — i.e., level L0 (see Appendix 4 for details) — disciplines $d$ of the articles published in a given journal $j$ in a given year $t$. Since a single article may cover more than one discipline, indicating the integration of knowledge and methods from different fields of study, we compute a fractional discipline count for every article, sum across all articles published in the journal in the specified year, and divide by the total number of articles so as to get the probability. We use $p_{jt}(d)$ to define an interdisciplinarity index $I_d(j,t)$ by making use of Shannon's entropy (*Shannon, 1948*), which captures the diversity of disciplinary representation in the journal's content:

$$I_d(j,t) = -\sum_d p_{jt}(d) \, \log_2\left(p_{jt}(d)\right) \, . \tag{1}$$

A higher index value indicates greater diversity in the disciplinary focus of a journal, while a lower index value suggests a more concentrated disciplinary focus. *Appendix 6—figure 1* demonstrates that higher values of $I_d$ are due to more papers drawing from multiple disciplines than to single-discipline papers from multiple disciplines being published in the journal.

Similarly, to quantify the internationalization of a journal, we make use of the fact that we can calculate the probability distribution $p_{jt}(n)$ of the countries $n$ of affiliation of the authors of the articles published in a given journal $j$ in a given year $t$. Since a single article may include affiliations from more than one country, we calculate a fractional country count for every article, sum across all articles published in the journal in the specified year, and divide by the total number of articles so as to get the probability. We use $p_{jt}(n)$ to define an internationalization index $I_n(j,t)$ by making use of Shannon's entropy (*Shannon, 1948*):

$$I_n(j,t) = -\sum_n p_{jt}(n) \, \log_2\left(p_{jt}(n)\right) \, . \tag{2}$$

A higher index value indicates greater diversity in the countries in authors' affiliations from the articles published in a journal, while a lower index value suggests a more restricted set of countries. *Appendix 7—figure 1* suggests that higher values of $I_n$ are not necessarily due to more papers authored by researchers from multiple countries but can be due (see the case of Chemistry) to papers with authors from a single country but with the greater variety of the specific country.

Shannon's entropy is one of many measures widely used to quantify deviation from random expectation (*Porter and Rafols, 2009*, *Grupp, 1990*). Indeed, many researchers in scientometrics and science of science have focused on so-called Rao-Stirling indices (*Leydesdorff et al., 2019*), which can account for heterogeneous distances between disciplines and sub-disciplines. Because we are focusing on the highest level in the hierarchy for Field of Study, this is not as much a concern for us. Additionally, the raising of 2 to the index values provides an 'effective' number of different categories in the data, something that helps with gaining insight from the results.

## Acknowledgements

We appreciate the comments from Spencer Hong, Maalavika Pillai, Reese Richardson, Mengyi Sun, Helio Tejedor, and Feihong Xu for the first draft of this manuscript. We sincerely thank the editor and the two anonymous reviewers for their thoughtful and constructive comments. Thanks to Aaron Geller and Helio Tejedor for the suggestions on data visualizations.

## Additional information

### Funding
No external funding was received for this work.

### Author contributions
Huaxia Zhou, Conceptualization, Data curation, Software, Validation, Investigation, Visualization, Methodology, Writing – original draft, Writing – review and editing; Luis A Nunes Amaral, Conceptualization, Resources, Data curation, Supervision, Funding acquisition, Validation, Visualization, Methodology, Writing – original draft, Project administration, Writing – review and editing

### Author ORCIDs
Huaxia Zhou ⓘ https://orcid.org/0000-0003-2822-0091
Luis A Nunes Amaral ⓘ https://orcid.org/0000-0002-3762-789X

Reviewer #1 (Public review): https://doi.org/10.7554/eLife.107765.3.sa1
Reviewer #2 (Public review): https://doi.org/10.7554/eLife.107765.3.sa2
Author response https://doi.org/10.7554/eLife.107765.3.sa3

## Additional files

### Supplementary files
MDAR checklist

### Data availability
The latest version of the OpenAlex data snapshot is freely available for download at https://docs.openalex.org/download-all-data/download-to-your-machine. The version we used for this work is the one released in March 11, 2022. The information of the version is available https://github.com/ourresearch/openalex-guts (*Piwowar, 2022*). We selected journal articles published between 1900 and 2020 that included both authorship and Field of Study information, resulting in a total of 49,478,866 records. The Clarivate Analytics' Journal Citation Reports are available and downloaded at https://jcr.clarivate.com/jcr/browse-categories. We downloaded the Journal Citation Reports (JCR) data in 2022 and selected the journals in natural sciences, life sciences, social sciences, and multidisciplinary sciences. With the availability of ISSN or eISSN information, we linked these journals to their corresponding publications in OpenAlex. The code needed to reproduce these results is available for download at the GitHub repository: https://github.com/amarallab/Evolution_of_Scientific_Journals (copy archived at *huaxiaz, 2025*).

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

## Appendix 1

### Calculation of the doubling time

The doubling time serves as an indicator of the growth rate of scientific publications (**de Solla Price, 1986**) representing the time required for the total output to double in volume. The doubling time $T_d$ is given by:

$$T_d = \frac{\ln(2)}{\frac{\ln(V_t/V_0)}{t}} \tag{3}$$

where:

$V_0$: Initial value at $t_0$

$V_t$: Final value after time period $t$

$t$: Time period

$T_d$: Doubling time

The denominator is called the growth rate $r$ :

$$r = \frac{\ln\left(\dfrac{V_t}{V_0}\right)}{t} \tag{4}$$

Sometimes, the doubling time can also be written as

$$T_d = \frac{\ln(2)}{r} \tag{5}$$

## Appendix 2

### Statistics on journal publications at country level

Journal articles indexed in OpenAlex include information on authors and their corresponding affiliations. We extracted the countries of those affiliations and the total aggregate volume of journal articles published by each country from 1900–2020. The five countries with the most publications are the United States (the U.S.), China, the United Kingdom (the U.K.), Japan, and Germany. *Figure 1* shows their annual number of publications. The number of journal articles published by Japan surpassed those by Germany in 1970 and those by the U.K. in 1980. Since 2005, China has published more journal articles annually than Germany, Japan, or the U.K., suggesting its rising role in knowledge creation.

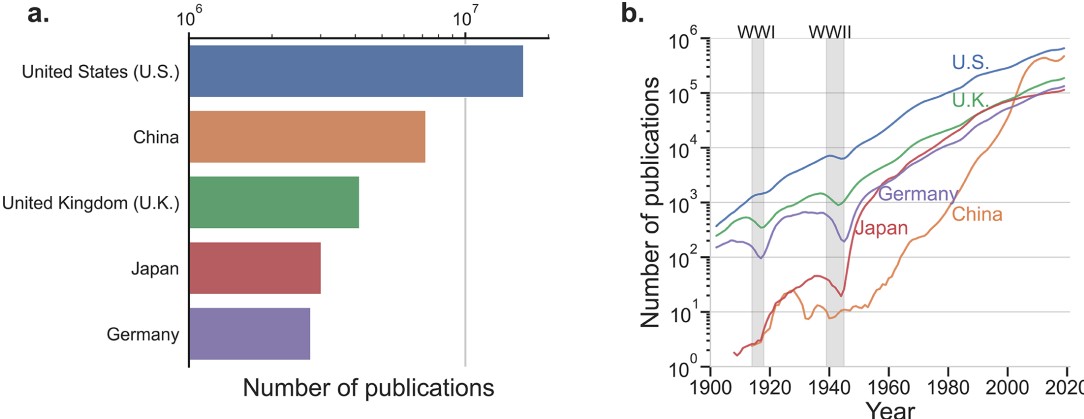

**Appendix 2—figure 1.** Journal article publication trends of the five most prolific countries. (**a**) Total number of journal publications (1900–2020) of the five most prolific countries. (**b**) Annual number of journal articles published by those five countries. For clarity, we show 5 year moving averages.

## Appendix 3

### Reliance on single data source

OpenAlex is far from comprehensive (especially for works published prior to 1960) and is not perfectly accurate. Thus, one may question how much results obtained using its data can be trusted. In order to answer this question, it is important to also acknowledge that these weaknesses are not unique to OpenAlex — they are shared by other bibliometric databases. For example, we observed that both the original Microsoft Academic Graph database (*Herrmannova and Knoth, 2016*) and the subscription-based Dimensions database (*Orduña-Malea and López-Cózar, 2018*) show similar patterns with respect to the fraction of records with missing information. Scopus (*Baas et al., 2020*), another popular database for bibliometric studies, does not have appropriate coverage for works published before 1996 (*Li et al., 2010*).

Because all datasets have similar weaknesses, but not all are equally available to all researchers, we chose to conduct our study using OpenAlex because its openness increases the reproducibility of our study. In addition, we focused our attention on what we believed would be the highest quality records. Specifically, we decided to focus on (i) complete records (i.e., those with authorship, affiliations, and field of study tags), and (ii) the most well-known, highest reputation journals.

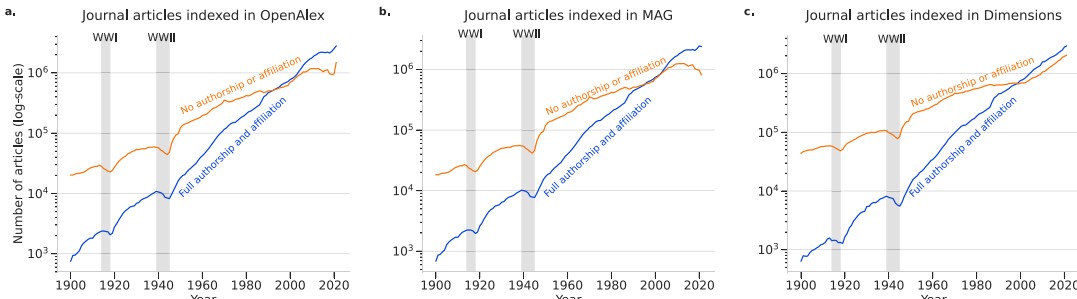

**Appendix 3—figure 1.** Number of journal articles over time across three bibliometric databases. (**a**) Number of journal articles indexed in OpenAlex over the past decade, categorized by the presence or absence of authorship and affiliation metadata. (**b**) Number of journal articles indexed in Microsoft Academic Graph (MAG) over the past decade, categorized by the presence or absence of authorship and affiliation metadata. (**c**) Number of journal articles indexed in Dimensions over the past decade, categorized by the presence or absence of authorship and affiliation metadata.

## Appendix 4

## Fields of study

Journal articles indexed in OpenAlex are assigned one or more topics called 'Field-of-Study (FoS)'. The FoS classification system is structured hierarchically across six levels (L0 up to L5). The snapshot adheres to the Microsoft Academic Graph (MAG) schema, wherein the classification process involves several stages: concept discovery (identifying relevant topics within the text), concept tagging (associating the identified topics with the corresponding papers), and hierarchy construction (organizing topics into a coherent structure). This process is largely automated, relying on web crawling, scraping, natural language processing, and machine learning algorithms to assign topics efficiently. Following automation, the outcomes were rigorously evaluated by a panel of human experts to ensure accuracy and relevance (*Shen et al., 2018*). The FoS classification has been widely applied in other research contexts to determine the disciplinary focus of journal articles (*Li et al., 2022*) and to build cumulative knowledge portfolio for scholars (*van der Wouden and Youn, 2023*).

We focus here on the top level (L0). The classifications at L0 levels contain 19 distinct fields representing traditional academic disciplines (e.g. Medicine, Biology and Chemistry). *Figure 1a* shows the total number of journal articles published for each L0 field; *Figure 1b-f* displays the number of journal articles published annually in each respective L0 field (from the largest to the smallest number of publications).

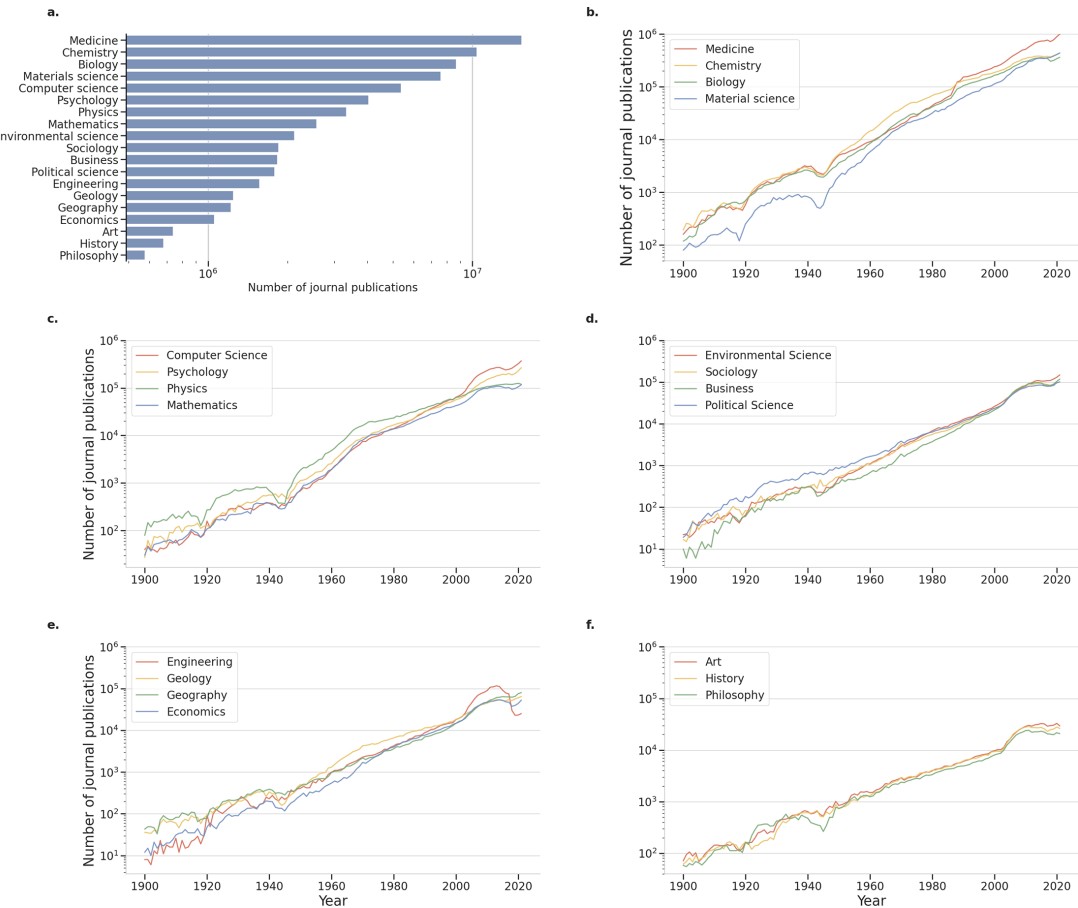

**Appendix 4—figure 1.** Number of journal publications in each Field of Study (FoS). (**a**) Total number of journal publications (1900–2021) in each L0 FoS. (**b-f**) Annual publication count of each L0 FoS.

## Appendix 5

### Journal selection

We focus on journals classified by the Clarivate Analytics' Science Citation Index Expanded (SCIE) into the following five categories: Multidisciplinary Sciences (73 journals in total), Biochemistry and Molecular Biology (285 journals in total), Multidisciplinary Chemistry (although the term used in the SCIE is Multidisciplinary Chemistry, these journals are considered as Chemistry journals; 175 journals in total), Multidisciplinary Physics (although the term used in the SCIE is Multidisciplinary Physics, these journals are considered as Physics journals; 84 journals in total), and General and Internal Medicine (167 journals in total).

We also focus on journals classified by the Clarivate Analytics' Social Science Citation Index (SSCI) into the following two categories: Economics (381 journals in total) and Political Science (187 journals in total).

We selected journals in each category based on the length of publication history, the impact of publications, and the volume of publications. We list the journals selected below along with their Journal Citation Report (JCR) abbreviations in parentheses, unless the JCR abbreviation matches the title of the journal (journals in each category are listed in alphabetical order).

### Multidisciplinary sciences

We selected the following seven journals:

> *Nature*
> *Nature Communications* (*Nat Commun*)
> *PLOS One - Proceedings of National Academy of Sciences* (*PNAS*)
> *Science - Science Advances* (*Sci Adv*)
> *Scientific Reports* (*Sci Rep-UK*)

The average Journal Impact Factor of these seven journals is 24.5, the average publication volume is 114,043 articles, and the average age is approximately 56 years.

### Multidisciplinary chemistry

We selected the following 16 journals:

> *ACS Nano*
> *Advanced Functional Materials* (*Adv Funct Mater*)
> *Advanced Materials* (*Adv Mater*)
> *Angewandte Chemie International Edition* (*Angew Chem Int Edit*)
> *Archiv der Pharmazie* (*Arch Pharm*)
> *Bioconjugate Chemistry* (*Bioconjugate Chem*)
> *ChemSusChem - Green Chemistry* (*Green Chem*)
> *Journal of the American Chemical Society* (*J Am Chem Soc*)
> *Journal of Controlled Release* (*J Control Releas*)
> *Journal of Physics and Chemistry of Solids* (*J Phys Chem Solids*)
> *Lab on a Chip* (*Lab Chip*)
> *Langmuir*
> *Nano Letters* (*Nano Lett*)
> *Small*
> *Ultrasonics Sonochemistry* (*Ultrason Sonochem*)

The average Journal Impact Factor in 2022 of these 16 journals is 11.4, the average publication volume is 22,309 articles, and the average age is approximately 45 years.

### Multidisciplinary physics

We selected the following 16 journals:

> *Annalen der Physik* (*Ann Phys-Berlin*)
> *Annals of Physics* (*Ann Phys-New York*)

*Chaos Solitons and Fractals (Chaos Soliton Fract)*
*Classical and Quantum Gravity (Classical Quant Grav)*
*Europhysics Letters (EPL-Europhys Lett)*
*European Physical Journal-Special Topics (Eur Phys J-Spec Top)*
*Foundations of Physics (Found Phys)*
*General Relativity and Gravitation (Gen Relat Gravit)*
*International Journal of Theoretical Physics (Int J Theor Phys)*
*Journal of Physics A-Mathematical and Theoretical (J Phys A-Math Theor)*
*Nature Physics (Nat Phys) - New Journal of Physics (New J Phys)*
*Physical Review Letters (Phys Rev Lett)*
*Physica Scripta (Phys Scripta)*
*Quantum Information Processing (Quantum Inf Process)*
*Soft Matter*

The average Journal Impact Factor in 2022 of these 16 journals is 4.3, the average publication volume is 13,981 articles, and the average age is approximately 44 years.

## Biochemistry and molecular biology

We selected the following 16 journals:

*American Journal of Respiratory Cell and Molecular Biology (Am J Resp Cell Mol)*
*Biomacromolecules - Cell - Cellular and Molecular Life Sciences (Cell Mol Life Sci)*
*Current Biology (Curr Biol) - EMBO Journal (EMBO J)*
*Free Radical Biology and Medicine (Free Radical Bio Med)*
*Genome Research*
*International Journal of Biological Macromolecules (Int J Biol Macromol)*
*Journal of Lipid Research (J Lipid Res)*
*Molecular Biology and Evolution (Mol Biol Evol)*
*Molecular Cell (Mol Cell)*
*Nature Structural and Molecular Biology (Nat Struct Mol Biol)*
*Nucleic Acids Research (Nucleic Acids Res)*
*Plant Cell*

The average Journal Impact Factor in 2022 of these 16 journals is 17.4, the average publication volume is 15,879 articles, and the average age is approximately 42 years.

## General and internal medicine

We selected the following 16 journals:

*American Journal of Medicine (Am J Med)*
*American Journal of Preventive Medicine (Am J Prev Med)*
*Annals of Internal Medicine (Ann Intern Med)*
*British Journal of General Practice (Brit J Gen Pract)*
*Chinese Medical Journal (Chinese Med J-Peking)*
*European Journal of Clinical Investigation (Eur J Clin Invest)*
*JAMA - J Am Med Assoc*
*JAMA Internal Medicine*
*Journal of General Internal Medicine (J Gen Intern Med)*
*Journal of the Royal Society of Medicine (J Roy Soc Med)*
*Lancet*
*Medical Clinics of North America (Med Clin N Am)*
*Medical Journal of Australia (Med J Australia)*
*Nature Medicine (Nat Med)*
*New England Journal of Medicine (New Engl J Med)*
*Preventive Medicine*

The average Journal Impact Factor in 2022 of these 16 journals is 39.1, the average publication volume is 16,159 articles, and the average age is approximately 88 years.

## Economics

We selected the following 16 journals:

> *American Economic Review* (*Am Econ Rev*)
> *Ecological Economics* (*Ecol Econ*)
> *Economic Geography* (*Econ Geogr*)
> *Economic Modelling* (*Econ Model*)
> *Energy Economics* (*Energ Econ*)
> *International Journal of Forecasting* (*Int J Forecasting*)
> *Journal of Development Economics* (*J Dev Econ*)
> *Journal of Econometrics* (*J Econometrics*)
> *Journal of Monetary Economics* (*J Monetary Econ*)
> *Journal of Public Economics* (*J Public Econ*)
> *Journal of Urban Economics* (*J Urban Econ*)
> *Quarterly Journal of Economics* (*Q J Econ*)
> *Review of Economic Studies* (*Rev Econ Stud*)
> *Review of Financial Studies* (*Rev Financ Stud*)
> *Small Business Economics* (*Small Bus Econ*)
> *World Development* (*World Dev*)

The average Journal Impact Factor in 2022 of these 16 journals is 7.9, the average publication volume is 3,662 articles, and the average age is approximately 55 years.

## Political science

We selected the following 16 journals:

> *American Journal of Political Science* (*Am J Polit Sci*)
> *American Political Science Review* (*Am Polit Sci Rev*)
> *Canadian Journal of Political Science* (*Can J Polit Sci*)
> *Comparative Political Studies* (*Comp Polit Stud*)
> *European Journal of Political Research* (*Eur J Polit Res*)
> *International Studies Review* (*Int Stud Rev*)
> *Journal of Conflict Resolution* (*J Conflict Resolut*)
> *Journal of Peace Research* (*J Peace Res*)
> *Journal of Politics* (*J Polit*)
> *Policy Studies Journal* (*Policy Stud J*)
> *Political Geography* (*Polit Geogr*)
> *Political Psychology* (*Polit Psychol*)
> *Political Studies* (*Polit Stud-London*)
> *PS: Political Science and Politics* (*PS-Polit Sci Polit*)
> *Public Administration* (*Public Admin*)
> *Public Opinion Quarterly* (*Public Opin Quart*)

The average Journal Impact Factor in 2022 of these 16 journals is 4.0, the average publication volume is 3,338 articles, and the average age is approximately 60 years.

## Appendix 6

### Relation between interdisciplinarity index and fraction of papers assigned multiple disciplines

We calculate the ratio between the number $n_m$ of papers with multiple disciplines (or FoS tags) and the number $n_s$ of papers with a single discipline (or FoS tag) for disciplinary journals in Chemistry, Physics, Biology, Medicine, Economics, and Political Science. We then plot the ratio versus the 2021 Interdisciplinarity Index of the journal. A positive slope in these plots indicated that greater values of $I_d$ correspond to higher proportions of papers drawing from multiple disciplines.

### Interdisciplinarity dynamics

The marginal distributions of Interdisciplinarity Index and 5-year trend of 16 disciplinary journals in Chemistry, Physics, Biology, Medicine, Economics, Political Science and 7 multidisciplinary journals.

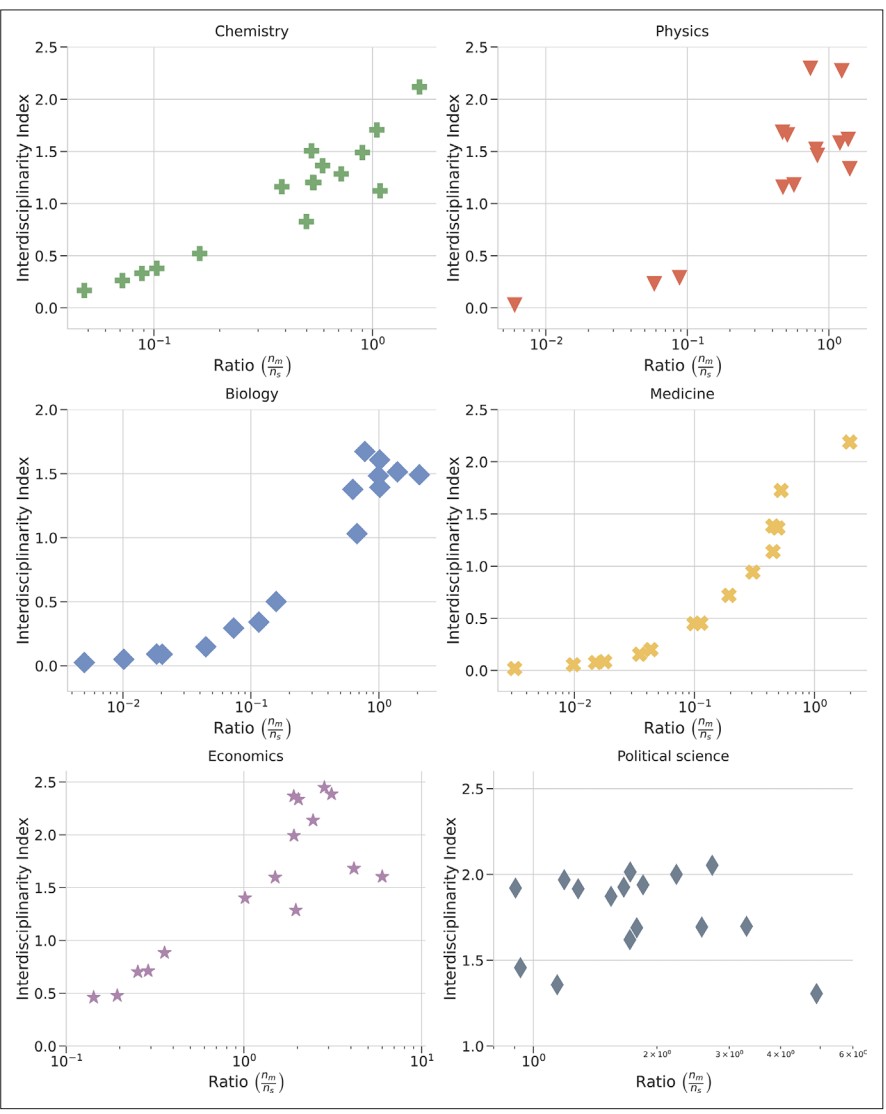

**Appendix 6—figure 1.** Relationship between the 2021 Interdisciplinarity Index and the ratio ($\frac{n_m}{n_s}$) for disciplinary journals in Chemistry, Physics, Biology, Medicine, Economics, and Political Science. Two Physics journals — *Annals of Physics* and *General Relativity and Gravitation* — and one journal in Medicine, *Medical Clinics of North America*, have ratio values equal to zero, meaning they do not appear in the semi-logarithmic plot. For Political Science, the range of both variables is notably narrow.

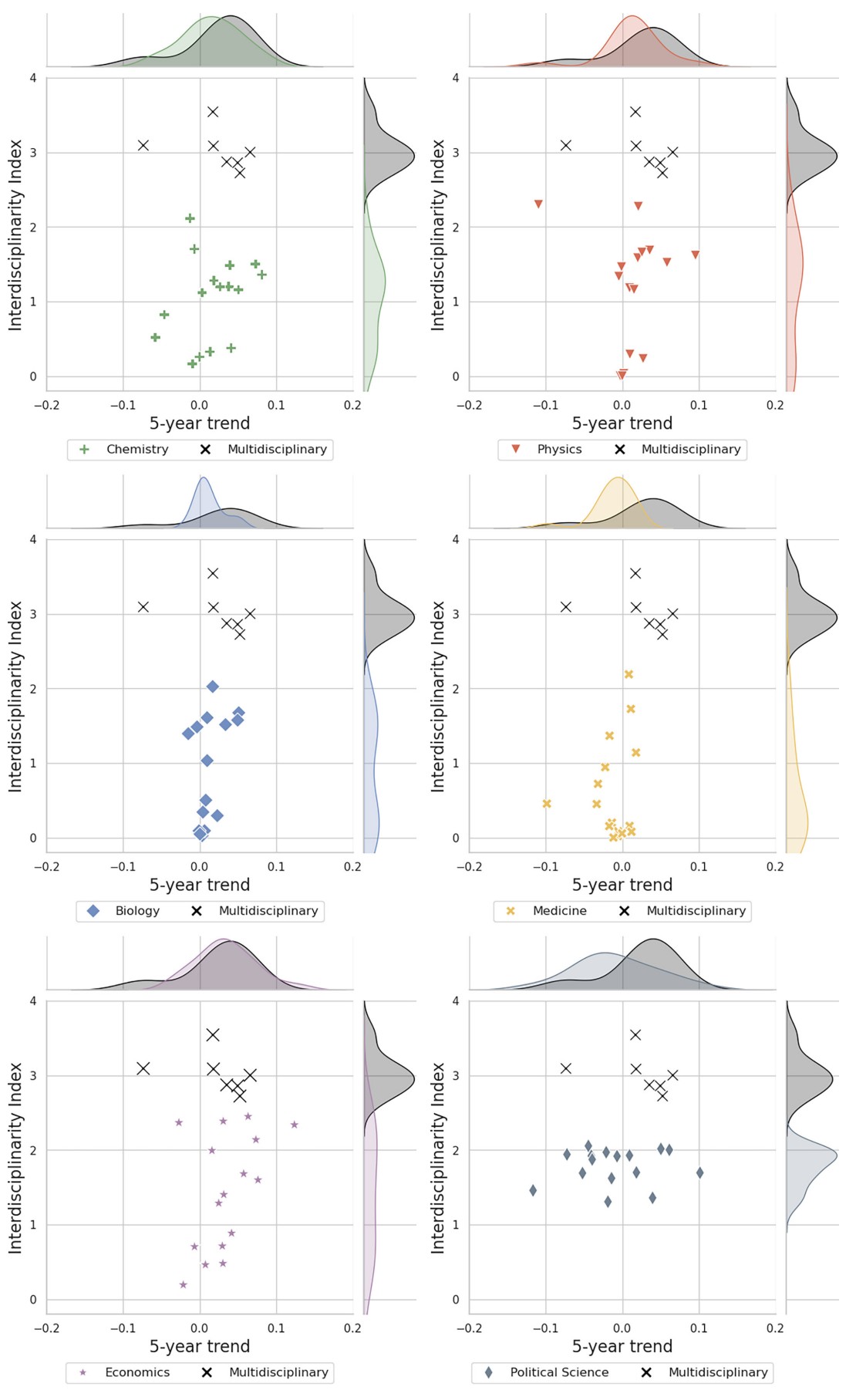

**Appendix 6—figure 2.** The marginal distribution of the Interdisciplinarity Index and the 5 year trend of 16 disciplinary journals compared to seven multidisciplinary journals for 2020.

## Appendix 7

### Relation between internationalization index and the fraction of papers with authors with affiliations in multiple countries

We calculate the ratio between the number $n_m$ of papers with authors affiliated to institutions from multiple countries and the number $n_s$ of papers with authors affiliated to institutions in a single country for disciplinary journals in Chemistry, Physics, Biology, Medicine, Economics, and Political Science. We then plot the ratio versus the 2021 Internationalization Index of the journal. A positive slope in these plots indicated that the greater values of $I_n$ correspond to higher proportions of papers drawing from authors from multiple countries.

### Internationalization dynamics

The marginal distributions of Internationalization Index and 5-year trend of 16 disciplinary journals in Chemistry, Physics, Biology, Medicine, Economics, Political Science and seven multidisciplinary journals.

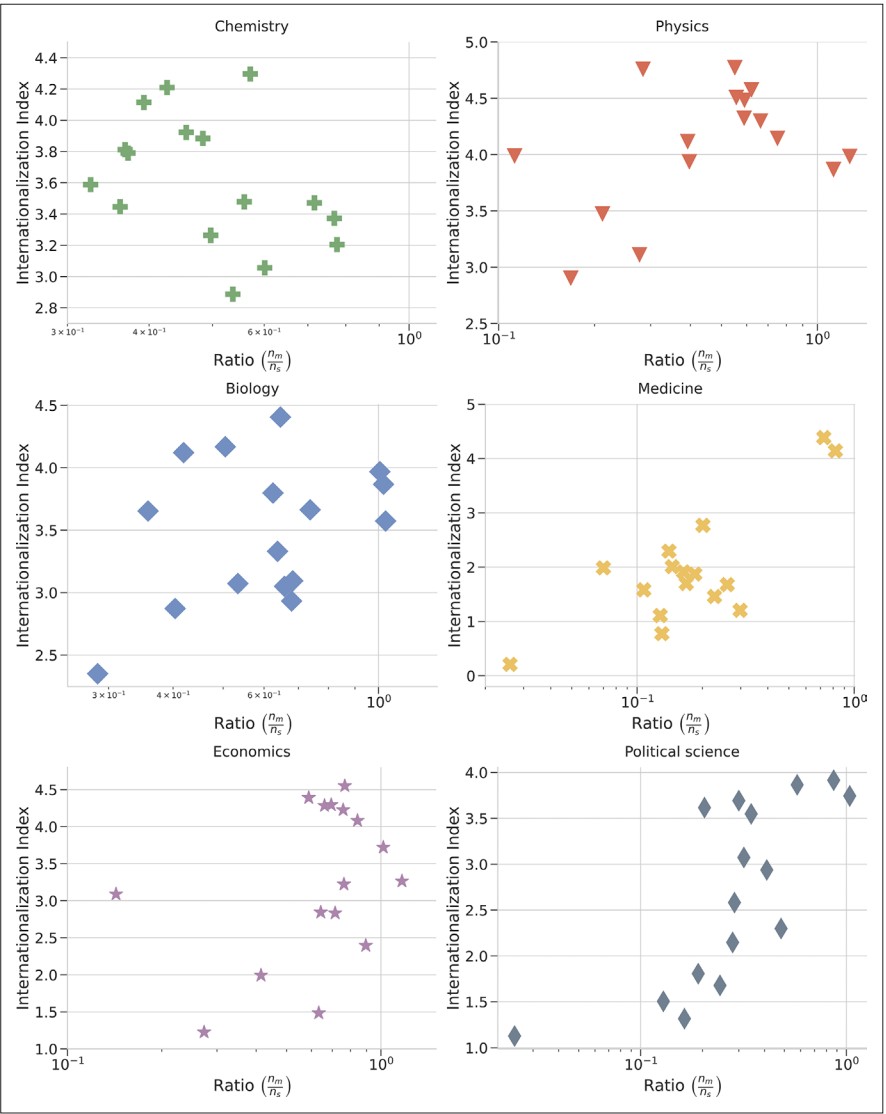

**Appendix 7—figure 1.** Relationship between the 2021 Internationalization Index and the ratio $\left(\frac{n_m}{n_s}\right)$ for disciplinary journals in Chemistry, Physics, Biology, Medicine, Economics, and Political Science. Notice that the range of values of the ratio is in many cases quite limited and close to 1, unlike the findings for interdisciplinarity. Moreover, correlation is not always positive.

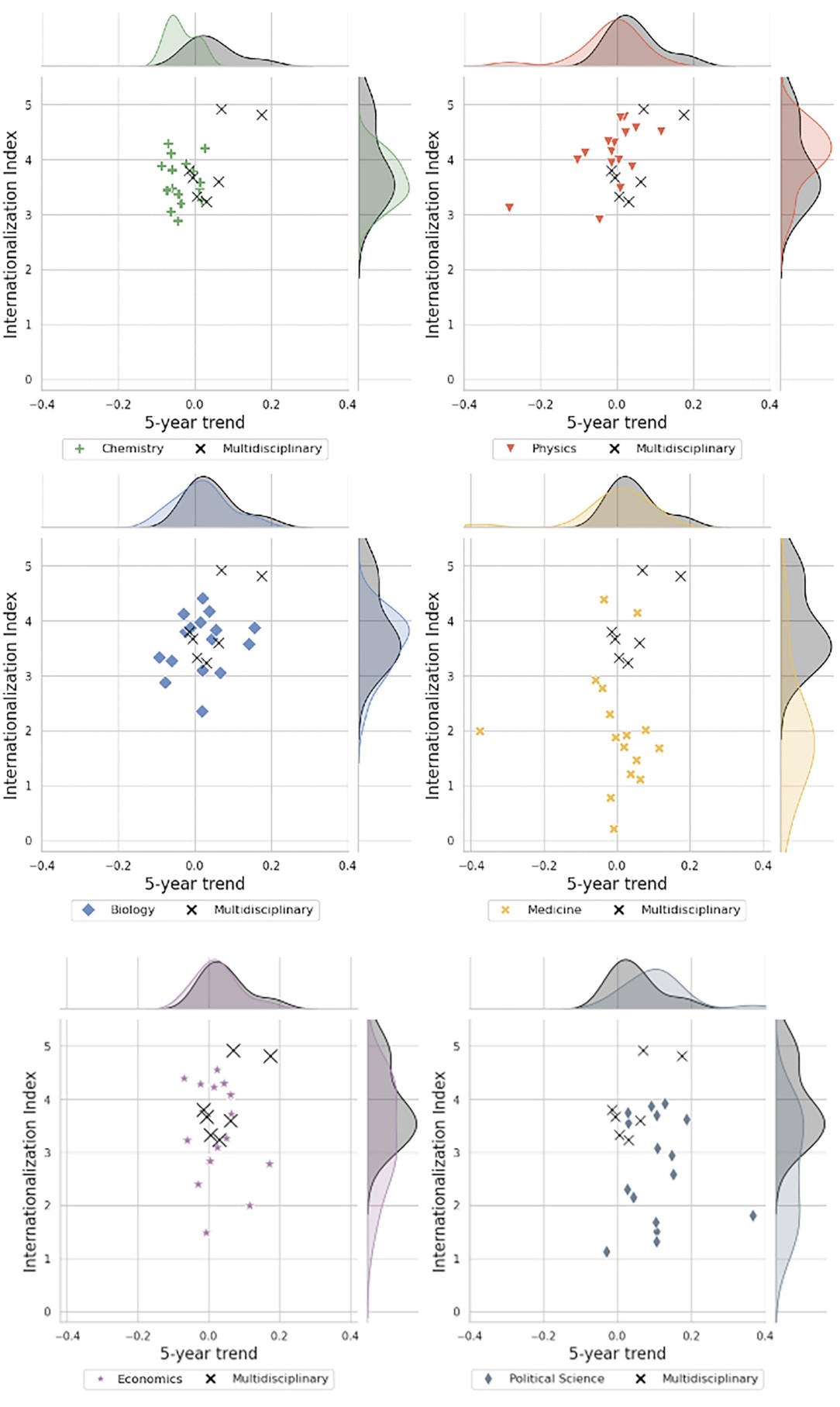

**Appendix 7—figure 2.** The marginal distribution of the Internationalization Index and the 5 year trend of 16 disciplinary journals to seven multidisciplinary journals for 2020.

## Appendix 8

### Team size

We investigated whether our findings for those three disciplines (Biology, Physics, and Medicine) could have been due to the characteristics of the largest teams. As a control, we repeated the previous analyses for these three disciplines but excluding all publications with 10 or more authors.

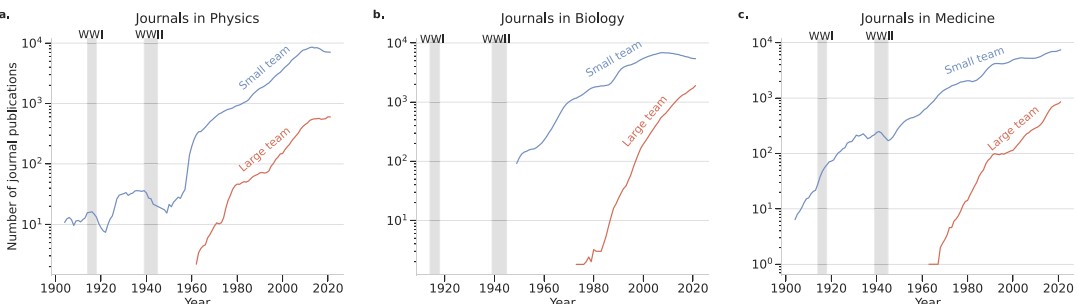

**Appendix 8—figure 1.** Team sizes in journal articles are notably large in the fields of Physics, Biology, and Medicine. The maximum number of authors in a Physics journal article is 4473, while Biology and Medicine journals have maximum author counts of 676 and 798, respectively. We define large teams as those with 10 or more authors and small teams as those with fewer than 10 authors across all three disciplines. The 5 year moving average of articles with large teams began to increase across all three fields after 1960.

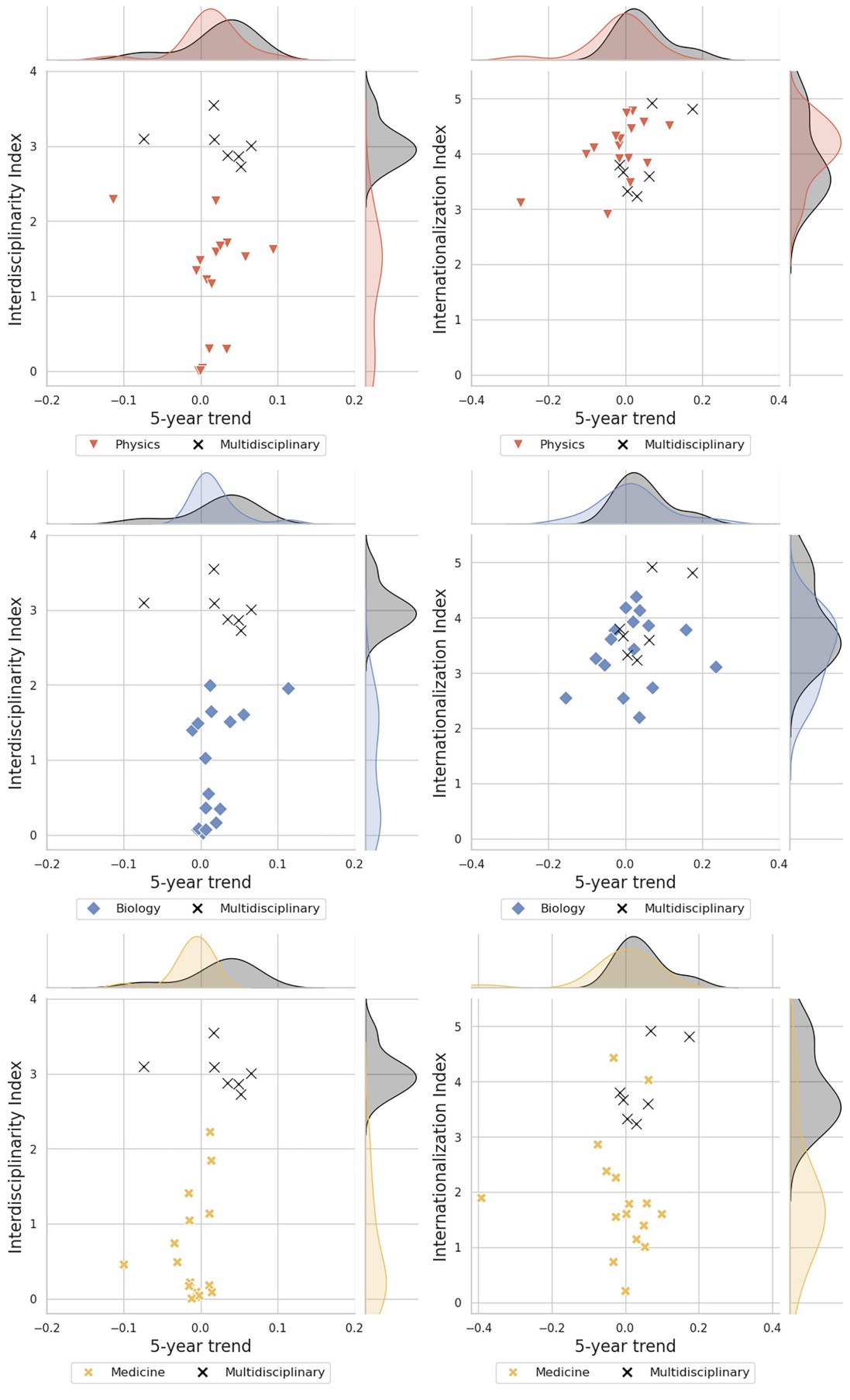

**Appendix 8—figure 2.** The marginal distribution of the Interdisciplinarity Index (Left) and Internationalization Index (Right) and the 5 year trend of 16 journals in Physics, Biology or Medicine, excluding the articles with 10 and more authors compared to seven multidisciplinary journals.

## Appendix 9

### Index temporal trend

We used two-dimensional plots showing the value of the Index for a given journal in a given year against the 5 year trend in the Index as a way to visualize the paths toward interdisciplinarity or internationalization followed by different disciplines.

In order to clarify exactly how the 5 year trend is calculated, consider the data in *Figure 1*. The figure shows the Interdisciplinary Index for two physics journals between 2005 and 2010, inclusive. The 5 year trend is defined as the linear coefficient of an ordinary least squares linear fit to the 6 data points for the Index in years 2005–2010.

The data suggest the reasons why we consider 5-year trends instead of other values, such as 3-, 10-, or 20-year trends. Smaller time windows would yield more uncertain trend estimates, making comparisons across years or journals less reliable. Larger time windows would potentially aggregate over changes in the trend.

The fact that reasonable values for the size of the time window are constrained between 4 and 7 years precludes the need for a sensitivity analysis of the window size.

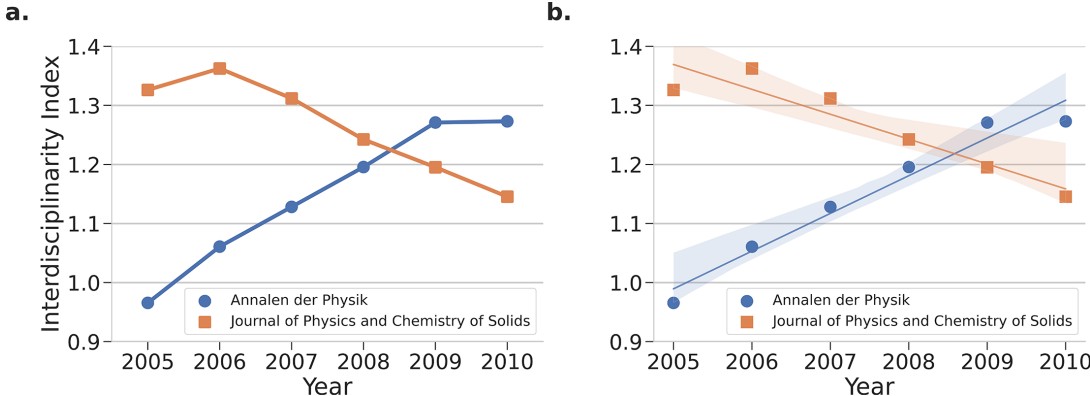

**Appendix 9—figure 1.** The Interdisciplinarity Index ($I_d$) of *Annalen der Physik* and *Journal of Physics and Chemistry of Solids* from 2005 to 2010. (**a**) Time evolution of $I_d$ for the two journals from 2005–2010. (**b**) The best-fit regression lines for $I_d$ over the same period with 95% confidence intervals (shaded regions).

