## [Editor Report · eLife Assessment]

This **important** study uses data from OpenAlex on more than 50 million journal articles in over 50,000 research journals to examine the dynamics of interdisciplinarity and international collaboration in research journals. The data analytics used to quantify disciplinary and national diversity are **convincing**, and support the claims that journals have become more diverse in both aspects. The revisions made by the authors have addressed the small number of concerns the reviewers had about the original version.

---

## [Referee Report · Reviewer #1 (Public review)]

(1) Summary

The authors aim to explore how interdisciplinarity and internationalization-two increasingly prominent characteristics of scientific publishing-have evolved over the past century. By constructing entropy-based indices from a large-scale bibliometric dataset (OpenAlex), they examine both long-term trends and recent dynamics in these two dimensions across a selection of leading disciplinary and multidisciplinary journals. Their goal is to identify field-specific patterns and structural shifts that can inform our understanding of how science has become more globally collaborative and intellectually integrated.

(2) Strengths

The primary strengths of the paper remain its comprehensive temporal scope and use of a rich, openly available dataset covering over 56 million articles. The interdisciplinary and internationalization indices are well-founded and allow meaningful comparisons across fields and time. The revised manuscript has substantially improved in several aspects. In particular, the authors have clarified the methodology of trend estimation with a concrete example and justification of the 5-year window, making their approach much more transparent. They have also expanded the discussion of potential disparities in data coverage across disciplines and time, acknowledging limitations and implementing safeguards in their analysis. Furthermore, the manuscript has been carefully revised for grammar, clarity, and style, which improves its overall polish. While a sensitivity analysis might still further strengthen the robustness of findings, the revisions satisfactorily address the main methodological concerns raised in the initial review.

(3) Evaluation of Findings

The findings, such as the sharp rise in internationalization in fields like Physics and Biology, and the divergence in interdisciplinarity trends across disciplines, are clearly presented and better substantiated in the revised version. The authors now provide more discipline-specific discussion (e.g., medicine, biology, social sciences), which adds valuable nuance to the interpretation of internationalization dynamics. The improved methodological clarity and acknowledgment of data limitations enhance the credibility of the results and their generalizability.

(4) Impact and Relevance

This study continues to make a timely and meaningful contribution to scientometrics, sociology of science, and science policy. Its combination of scale, historical depth, and field-level comparison offers a useful framework for understanding changes in scientific publishing practices. The entropy-based indicators remain a simple yet flexible tool, and the expanded discussion of their appropriateness strengthens the methodological foundation. The use of open bibliometric data enhances reproducibility and accessibility for future research. Policymakers, journal editors, and researchers interested in publication dynamics will likely find this work informative, and its methods could be applied or extended to other structural dimensions of scholarly communication.

---

## [Referee Report · Reviewer #2 (Public review)]

Summary:

This paper uses large-scale publication data to examine the dynamics of interdisciplinarity and international collaborations in research journals. The main finding is that interdisciplinarity and internationalism have been increasing over the past decades, especially in prestigious general science journals.

Strengths:

The paper uses a state-of-the-art large-scale publication database to examine the dynamics of interdisciplinarity and internationalism. The analyses span over a century and in major scientific fields in natural sciences, engineering, and social sciences. The study is well designed and has provided a range of robustness tests to enhance the main findings. The writing is clear and well organized.

---

## [Author Response]

The following is the authors’ response to the original reviews.

**Reviewer #1 (Public review):**
However, some methodological choices, such as the use of a 5-year sliding window to compute trend values, are insufficiently justified and under-explained. The paper also does not fully address disparities in data coverage across disciplines and time, which may affect the reliability of historical comparisons. Finally, minor issues in grammar and clarity reduce the overall polish of the manuscript.

We thank the reviewer for pointing out the weakness of the manuscript. We addressed these comments in our response to Recommendations A and B. Minor grammar and clarity issues have also been addressed.

**Reviewer #2 (Public review):**
The first thing that comes to mind is the epistemic mechanism of the study. Why should there be a joint discussion combining internationalism and interdisciplinarity? While internationalism is the tendency to form multinational research teams to work on research projects, interdisciplinarity refers to the scope and focus of papers that draw inspiration from multiple fields. These concepts may both fall into the realm of diversity, but it remains unclear if there is any conceptual interplay that underlies the dynamics of their increase in research journals.

We thank the reviewer for pointing out the lack of clarity in our decision to conduct a joint discussion of interdisciplinarity and internationalization.

It is a well-known fact that team science has increased in importance over time. An important question then is whether teams have only grown in size and frequency or whether they have changed in other aspects. Interdisciplinarity and internationalization are two aspects in which teams could have changed.

We revised the Introduction (Lines 68–70 of the revised manuscript) to address this matter.

It is also unclear why internationalization is increasing. Although the authors have provided a few prominent examples in physics, such as CERN and LAGO, which are complex and expensive experimental facilities that demand collective efforts and investments from the global scientific community, whether some similar concerns or factors drive the growth of internationalism in other fields remains unknown. I can imagine that these concerns do not always apply in many fields, and the authors need to come up with some case studies in diverse fields with some sociological theory to support their empirical findings.

We thank the reviewer for requesting further evidence concerning why our findings may be correct. Physics is an area where the need for extraordinary resources has naturally led to large international collaborative efforts. As we discuss in line 255 of the revised manuscript, this is actually also the case for biology. The Human Genome Project and subsequent projects have also required massive investments, leading to further internationalization.

We believe that the drive toward internationalization for medicine has to do with the need for establishment of robust results that are not specific to a single country or medical system. Additionally, the impact of global epidemics — Acquired immunodeficiency Syndrome (AIDS), Severe Acute Respiratory Syndrome (SARS) — has also increased the needs to involve researchers from around the world.

The case for increased internationalization in the social sciences is, we believe, related to the desire to identify phenomena that extend beyond the Western, educated, industrialized, rich and democratic (WEIRD) societies.

We have expanded the discussion around these points in lines 274–283 of the revised manuscript.

The authors use Shannon entropy as a measure of diversity for both internationalism and interdisciplinarity. However, entropy may fail to account for the uneven correlations between fields, and the range of value chances when the number of categories changes. The science of science and scientometrics community has proposed a range of diversity indicators, such as the RaoStirling index and its derivatives. One obvious advantage of the RS index is that it explicitly accounts for the heterogeneous connections between fields, and the value ranges from 0 to 1. Using more state-of-the-art metrics to quantify interdisciplinarity may help strengthen the data analytics.

We thank the reviewer for pointing the need to provide a deeper discussion of the impact of different metrics on how disciplinary diversity is calculated. We chose Shannon’s entropy because it accounts for both richness (the number of distinct fields) and evenness (the balance of representation across fields). While measures such as the Rao-Stirling index can be very useful when considering disciplines at different levels of aggregation, since to consider only level 0 Field-of-Study (FoS) tags, that problem is not as much a concern for our analysis.

We have added a further clarification in lines 145–151 of the revised manuscript.

**Reviewer #1 (Recommendations for the authors)**
Ambiguity in the Trend Calculation Methodology in Figure 4 and 5The manuscript uses a 5-year sliding window to calculate recent trends in interdisciplinarity (I_d_) and internationalization (I_n_), but the method is not clearly described. Could the authors clarify whether the trend is calculated by (1) performing linear regression on the index values over the past 5 years, (2) using the regression slope as the trend value, and (3) interpreting the sign and magnitude of the slope to indicate increasing, decreasing, or stable trends? Additionally, the rationale for choosing a 5-year window over other durations (e.g., 10 or 15 years) is not discussed. Given that different time windows could yield different insights, a brief justification or sensitivity check would strengthen the methodological transparency.

Thank you for pointing the lack of clarity in our description. In an attempt to increase clarity, we added a specific case study to illustrate the use of 5-year trend in the Supplementary Information: Estimation of tendency of the revised manuscript (Lines 691–704 of the revised manuscript).

Specifically, imagine we want to calculate the trend of the Interdisciplinarity Index for 2010 for Annalen der Physik. We would perform an ordinary least squares linear fit to the 6 data points for the Index in years 2005–2010.

The reason to focus on a 5-year window is two-fold. First, a longer time period would — as suggested by the data on Figure S10 — likely aggregate over multiple trends. Second, a shorter time period would result in too great an uncertainty in the estimation of the trend.

This is the reason why we did not implement a sensitivity analysis. Reasonable time windows that consider the two reasons expressed above would be too narrow to provide a worthwhile analysis.

Lack of Discussion on Temporal Coverage Disparities Across DisciplinesThe study spans publications from 1900 to 2021, but the completeness and representativeness of the data-especially in earlier decades-may differ significantly across disciplines. For instance, OpenAlex has limited coverage for publications before the mid-20th century, and disciplines such as Medicine and Political Science may have adopted journal-based publishing at different historical periods compared to Physics or Chemistry. These temporal disparities could bias cross-disciplinary comparisons of long-term trends in interdisciplinarity and internationalization. I recommend that the authors briefly discuss this limitation and, if possible, report when coverage becomes reliable for each discipline. A sensitivity analysis starting from a common baseline year (e.g., 1950 or 1970) could also help assess whether the observed disciplinary differences are driven in part by unequal temporal data availability.

We thank the reviewer for the requesting further clarification on this matter. We completely agree that “completeness and representativeness of the data – especially in earlier decades-may differ significantly across disciplines”. That is exactly the reason why we made the analyses choices described in the manuscript.

Indeed, we consider only three journals for the analysis of the entire 1900–2021 period. Those 3 journals, Nature, PNAS and Science are ones that we know to be well recorded.

When conducting the disciplinary analysis, we focus on the period 1960–2021. While we know that the coverage for the social sciences is less robust until the 1990s, we address this concern by implementing several safeguards:

Manual selection of representative journals in each discipline to ensured that their publications are well represented in OpenAlex.

Decade by decade analysis of interdisciplinarity and internationalization so that changes over time can be identified and potential issues with data coverage are restricted to only some aspects of the analysis.

We also acknowledge the potential coverage disparities in earlier years of the data source (Lines 319-326 of the revised manuscript).

The authors use both interdisciplinarity and multidisciplinarity. While these concepts offer similar definitions of diversity, it may help the reader if there is some explanation to clarify their subtle differences. (Reviewer #2)

It is a well-known fact that team science has increased in importance over time. An important question then is whether teams have only grown in size and frequency or whether they have changed in other aspects. Interdisciplinarity and internationalization are two aspects in which teams could have changed.

We revised the Introduction (Lines 68–70 of the revised manuscript) to address this matter.

Minor CommentsSeveral sentences(1) Line 11: The phrase “authors form multiple countries” contains a typographical error. The word “form” should be corrected to “from” so that the sentence reads: “authors from multiple countries.”tences and phrases throughout the manuscript could be improved for grammatical accuracy, clarity, and stylistic appropriateness:(2) Line 63: The clause “these expansion is well described by a logistic model” contains a subject-verb agreement error. “These” should be replaced by the singular demonstrative pronoun “this”, resulting in: “This expansion is well described by a logistic model.”(3) Line 89: The phrase “were quickly overcame” misuses the verb form. “Overcame” is a past tense form and should be replaced with the past participle “overcome” to match the passive construction. Suggested revision: “were quickly overcome.”(4) Line 106: The verb “refered” is misspelled. It should be corrected to “referred” for proper past tense. The corrected phrase should read: “we referred to...”(5) Line 127: The phrase “sing discipline papers” contains a typographical error. “Sing” should be “single”, yielding: “single discipline papers.”(6) Lines 238–239: The sentence “An exception to this pattern are the two mega open-access journals: PLOS One and Scientific Reports, which have internationalization indices as high the the most internationalized Physics journals.” contains multiple grammatical issues.First, the subject “An exception” is singular, but the verb “are” is plural; this results in a subject-verb agreement error.Second, the phrase “the the” includes a typographical repetition.Third, the comparative construction is incomplete; “as high the the...” is ungrammatical and should use “as high as.”Suggested revision: “An exception to this pattern is the pair of mega open-access journals— PLOS One and Scientific Reports—which have internationalization indices as high as those of the most internationalized Physics journals.”(7) Line 254: The sentence “biological research been revolutionized...” lacks an auxiliary verb. To be grammatically correct, it should read: “biological research has been revolutionized...”(8) Line 258: The phrase “need global spread of...” is syntactically awkward. Depending on the intended meaning, it could be revised to either “the global spread of...” or “the global need for the spread of...” for clarity.(9) Figure S2 Caption: The term “Microsofe Academic Graph” is a typographical error and should be corrected to “Microsoft Academic Graph.”(10) Reference [40]: The link “https://doi.org/10.1038/nature02168” is missing the “h” in “https.” The corrected version is: “https://doi.org/10.1038/nature02168.”

We appreciate your comments on the grammar and clarity of the manuscript. We have thoroughly reviewed and corrected these issues to improve the overall clarity of the text.

Line 11: We changed the typo “form” to “from”.

Line 63: We changed the sentence to “There has been a significant expansion in the number of countries where scientists are publishing in selective journals”.

Line 89 (Line 93 of the revised manuscript): We revised the sentence as suggested, and the revised sentence becomes “Even the significant impacts on publication rates of the two World Wars were quickly overcome, and exponential growth resumed. ”

Line 106 (Line 110 of the revised manuscript): We changed the typo “refered” to “referred”.

Line 127 (Line 131 of the revised manuscript): We changed the typo “Sing” to “single”.

Lines 238-239 (Lines 245-247 of the revised manuscript): We thank the issues pointed out by the reviewer, and we took the reviewer’s suggested version and changed the original sentence to “An exception to this pattern is the pair of mega open-access journals — PLOS One and Scientific Reports — which have internationalization indices as high as those of the most internationalized Physics journals”.

Line 254 (Line 262 of the revised manuscript): We added the auxiliary verb to the sentence, and the sentence now becomes “biological research has been revolutionized”

Line 258 (Line 266 of the revised manuscript): We changed the phrase to “the global need for the spread of”.

Figure S2 Caption: We corrected the typo of “Microsoft Academic Graph”.

Reference [40]: We corrected the URL of the reference.

**Reviewer #2 (Recommendations for author):**
Some typos:(1) Page 2: On page 2, “contributions from a multiple disciplines” and ”these expansion is well described”.(2) Page 4: “World Wars were quickly overcame”.(3) Page 5: “to quantify the the internationalization of a journal”.(4) Page 10: “indices as high the the most internationalized Physics journals”(5) Page 10: The sentence “indices as high the the most internationalized Physics journals” contains multiple issues. The phrase “the the” is a typographical error, and the comparative construction is incomplete. It should be revised to: “indices as high as those of the most internationalized Physics journals.”

We revised those typographical errors on page 2, 4, 5, and 10 pointed out by the reviewer. We truly thank the reviewer’s critical examination on the syntax of the manuscript.

Page 2: We removed “a” so now the sentence reads: “contributions from multiple disciplines.”

Page 2: We changed the sentence to “There has been a significant expansion in the number of countries where scientists are publishing in selective journals”.

Page 4: We replaced “overcame” with the past participle “overcome” , resulting in: “World Wars were quickly overcome.”

Page 5: The phrase “to quantify the the internationalization of a journal” contains a typographical repetition. We changed it to: “to quantify the internationalization of a journal.”

Page 10: For the sentence “indices as high the the most internationalized Physics journals”, we removed duplicated “the” as a typographical error. We revised the sentence into: “indices as high as those of the most internationalized Physics journals.”